# Estimating behavioural relaxation induced by COVID-19 vaccines in the first months of their rollout

Yuhan Li[1], Nicolò Gozzi[2], Nicola Perra[1,3]*

**1** School of Mathematical Sciences, Queen Mary University, London, United Kingdom, **2** ISI Foundation, Turin, Italy, **3** The Alan Turing Institute, London, United Kingdom

* n.perra@qmul.ac.uk

## Abstract

The initial rollout of COVID-19 vaccines has been challenged by logistical issues, limited availability of doses, scarce healthcare capacity, spotty acceptance, and the emergence of variants of concern. Non-pharmaceutical interventions (NPIs) have been critical to support these phases. However, vaccines may have prompted behavioural relaxation, potentially reducing NPIs adherence. Epidemic models have explored this phenomenon, but they have not been validated against data. Moreover, recent surveys provide conflicting results on the matter. The extent of behavioural relaxation induced by COVID-19 vaccines is still unclear. Here, we aim to study this phenomenon in four regions. We implement five realistic epidemic models which include age structure, multiple virus strains, NPIs, and vaccinations. One of the models acts as a baseline, while the others extend it including different behavioural relaxation mechanisms. First, we calibrate the baseline model and run counterfactual scenarios to quantify the impact of vaccinations and NPIs. Our results confirm the critical role of both in reducing infection and mortality rates. Second, using different metrics, we calibrate the behavioural models and compare them to each other and to the baseline. Including behavioural relaxation leads to a better fit of weekly deaths in three regions. However, the improvements are limited to a 2–10% reduction in weighted mean absolute percentage errors and these gains are generally offset by models' increased complexity. Overall, we do not find clear signs of behavioural relaxation induced by COVID-19 vaccines on weekly deaths. Furthermore, our results suggest that if this phenomenon occurred, it generally involved only a minority of the population. Our work contributes to the retrospective validation of epidemic models developed amid the COVID-19 Pandemic and underscores the issue of non-identifiability of complex social mechanisms.

**Data availability statement:** The data and code to replicate the results can be found in https://github.com/Jadecoool/Estimating-behavioural-relaxation-induced-by-COVID-19-vaccines-in-the-first-months-of-their-rollout.

## Author summary

The COVID-19 vaccines rollout initially faced significant challenges. Non-pharmaceutical interventions (NPIs) complemented vaccinations in addressing these

**Funding:** Y.L. acknowledges support from the China Scholarship Council (CSC). N.G. acknowledges support from the Lagrange Project of the Institute for Scientific Interchange Foundation (ISI Foundation) funded by Fondazione Cassa di Risparmio di Torino (Fondazione CRT). The funders had no role in study design, data collection and analysis, decision to publish, or preparation of the manuscript.

**Competing interests:** The authors have declared that no competing interests exist.

issues. However, the interplay between vaccines and NPIs is complex and still unclear. While some surveys suggest that the arrival of vaccines induced people to relax their protective behaviours, others found little support for this phenomenon. Furthermore, the epidemic models developed so far to study these processes lack empirical validation. We aim to quantify the extent of behavioural relaxation studying five compartmental models in four regions. All models integrate age-structure, multiple virus strains, NPIs, and vaccinations. Four also include vaccine induced behavioural relaxation mechanisms. Our findings confirm that both vaccinations and NPIs significantly reduced infection and mortality rates. Furthermore, although adding behavioural relaxation mechanisms improve the overall goodness of fit, the gains are limited and offset by increased models' complexity. Overall, we found little evidence of behavioural relaxation induced by vaccines on weekly deaths. Even if this phenomenon occurred, our results suggest that it generally involved only a minority of the population. Our work contributes to the efforts devoted to retrospectively validating epidemic models developed during the COVID-19 Pandemic and highlights the issue of non-identifiability of social mechanisms.

## Introduction

COVID-19 vaccines led to a significant reduction in mortality and transmission rates [1–5]. However, especially in the first months of their rollout, vaccination efforts have encountered many challenges due to logistical issues, insufficient stockpiles, limited healthcare capacity, and spotty acceptance [6]. A study from the United States, for instance, showed that counties with limited healthcare resources were also more likely to achieve lower COVID-19 vaccination rates [7]. On a global scale, initial vaccine acceptance varied significantly across different regions, ranging from 13% in Iraq to 97% in Vietnam according to surveys conducted before the start of vaccine rollout [8]. Moreover, vaccine nationalism (i.e., the prioritization of national self-interests over equitable global access) [9,10] and socioeconomic inequalities led to a concentration of doses in high-income countries [10–12]. The insufficient vaccination coverage in many areas proved inadequate to prevent subsequent waves and reduce both cases and deaths [5,13]. The vaccines rollout was also challenged by the emergence of SARS-CoV-2 variants of concern (VOC), such as Alpha and Delta, which led to a reduction of the protection against infection provided by vaccines [14,15].

While non-pharmaceutical interventions (NPIs) have been key to support vaccination efforts during these initial phases [4,5,16], their adoption is shaped by many factors such as perceived susceptibility, severity, barriers to actions, exposure to (mis)information, and peer-effects [17–23]. It is natural to wonder whether the arrival of COVID-19 vaccines impacted the adoption of NPIs. Indeed, the start of vaccinations might have lowered the risk perception in at least some groups of the population, which in turn might have led to a relaxation of compliance to a wide range of NPIs such as mask-wearing, social distancing, and increased hygiene practices. The potential effects of this phenomenon, which for simplicity we will refer to as *behavioural relaxation*, have been explored via epidemic models in realistic, yet theoretical, scenarios during the first months of the vaccine rollout [24–27]. The results from these efforts suggest that behavioural relaxation could reduce the positive gains brought about by vaccines thus leading to higher disease burden. The empirical evidence does not provide a clear picture of the extent of behavioural relaxation. Indeed, longitudinal survey data from 16 European countries suggest that vaccinated individuals had 1.31 times more social contacts than non-vaccinated [28]. Other surveys conducted in England and Wales report that, after two doses, about 48% of the respondents increased their interactions with friends or

family and about 38% visited indoor places more often [29]. A regression analysis conducted considering different mobility data in London shows a positive association between mobility and vaccinations [30]. Other surveys conducted in Brazil, Italy, South Africa, and the United Kingdom, indicate that public transport usage increased by up to 10% after the rollout of first dose of vaccine [31]. However, the results from large, and repeated, cross-sectional surveys conducted in France provide limited support for a systematic behavioural relaxation, especially during the initial phases of the vaccine rollout [32].

In this context, we investigate the interplay between vaccinations and NPIs during the first months of the COVID-19 vaccine rollout in four regions: British Columbia (Canada), Lombardy (Italy), London (United Kingdom), and São Paulo (Brazil). These regions have been selected to sample different epidemiological, socioeconomic, and socio-demographic contexts, as well as different vaccine rollout schedules and coverages. We set to quantify the extent of behavioural relaxations induced by the start of vaccination campaigns and estimate their potential impact on reported weekly deaths via epidemic models. Indeed, as mentioned above, the models published so far to capture such behavioural relaxation have not been validated against real data [24–27]. Besides, they have not been compared among them nor against simpler models that do not feature explicit behavioural relaxation mechanisms. By using the data collected and made available over the last years, we can address these gaps. To this end, we develop a series of stochastic compartmental epidemic models, integrating vaccinations, variants of concern, age-structure, NPIs, as well as individuals' behavioural relaxation linked to vaccines. In particular, we consider a baseline model without behavioural relaxation mechanisms and four models that include them. In these, we introduce explicit compartments that account for non-compliant individuals who relaxed their protective behaviours as a result of the start of vaccinations. These models, which we will refer to as *behavioural models*, differ according to the mechanisms used to describe the transitions in and out of non-compliant behavioural compartments.

To set the stage, we first calibrate the baseline model to reported data (i.e., weekly deaths) in the four locations and run two counterfactual scenarios to quantify the impact of vaccines and of NPIs on COVID-19 burden. Our results clearly confirm the crucial role played by both in reducing deaths and infections in the regions considered. Then, we calibrate and compare the other four models against each other and the baseline. Our results do not provide strong support for the inclusion of behavioural relaxation mechanisms across all regions. Even more, the phase space selected in the calibration suggests that, if behavioural relaxation took place, it was limited to a minority of the population thus not leaving clear marks on weekly reported COVID-19 deaths. Moreover, our findings highlight the issue of non-identifiability of complex behavioural dynamics in epidemic models.

## Results

We implement and compare five epidemic models. The first acts as a baseline. The others build on it and include different behavioural relaxation mechanisms. The four behavioural models combine and extend approaches from the literature [24,25]. To explore different epidemiological, socioeconomic and socio-demographic contexts, as well as different vaccine rollout schedules and coverages we consider four regions of the world: British Columbia (Canada), Lombardy (Italy), London (United Kingdom), and São Paulo (Brazil). All models are calibrated to confirmed weekly deaths via an Approximate Bayesian Computation-Sequential Monte Carlo (ABC-SMC) method [33]. While we provide a summary of the models in the next two sections, we refer the reader to the Material and Methods as well as in S1 Text (Sect 1.1) for full details.

## Baseline model

The baseline model (*baseline*) is a Susceptible-Latent-Infected-Recovered (SLIR) epidemic model integrating vaccinations, NPIs, age-structured contact matrices, multiple virus strains, and disease-related deaths. It constitutes the core upon which the other four models are built. We include age-stratified vaccinations by using real data [34–38]. For simplicity, we assume a single dose regiment and ignore the time required to develop full protection after inoculation. Furthermore, we assume that only susceptible individuals are eligible for vaccination. We estimate the impact of NPIs on social contacts using mobility data from the COVID-19 Community Mobility Report published by Google LLC [39] and the Oxford Coronavirus Government Response Tracker (OxCGRT) [40]. This data is used to modulate the contact matrices as function of time. We also consider the emergence and spread of a second virus strain. According to virological surveillance data, during the period under consideration, the Alpha variant emerged and replaced the ancestral type in British Columbia and Lombardy, while Delta replaced the Gamma VOC in São Paulo. London, during the time interval under investigation, faced primarily a wave dominated by Alpha [41–43]. In our models, we assume Alpha and Delta to have higher transmissibility [44,45], vaccine-induced immunity escape potential [44,46,47], and shorter latent period with respect to previously circulating strains [48–52]. We refer the reader to the Materials and Methods for more details.

## Behavioural relaxation models

Building on the baseline and the literature, we explore four different behavioural relaxation models. To this end, we extend the compartmental structure of the baseline by introducing non-compliant (NC) compartments to account for susceptible individuals who relax their COVID-19 safe behaviours as result of the vaccine rollout. The models differ for two main aspects (see Table 1 for a summary of models' characteristics). The first revolves on which groups of the population might relax their behaviours. Following Ref. [24], in the first and second behavioural model we assume that all susceptible individuals, independently from their vaccination status, might enter/leave the NC compartments. Following Ref. [25] instead, in the third and fourth behavioural models only susceptible and vaccinated individuals might transition to non-compliant compartments. The second aspect that further differentiate the models relates to the mechanisms regulating the transitions from and to compliance [24,25]. The first and third model consider the simplest transitions type: a constant transition rate. The second and fourth models consider transition rates that are time-dependent and function of the fraction of the population in specific compartments. In details, in the first and third model the transition from and to compliance takes place at constant rates $\alpha$ and $\gamma$, respectively. In the second and fourth models, the transition rate from compliant to non-compliant behaviour is set as a function of $\alpha$ and the fraction of vaccinated individuals. The reverse transition rate is set as a function of $\gamma$ and the number of daily deaths per 100,000. Indeed, daily deaths have been often used, especially by media, to characterize the severity of different Pandemic phases and are a known driver of individuals' adherence to NPIs [17]. To highlight the two key aspects differentiating the models, we refer to the first model as the *constant rate model*, the second as the *time-varying rate model*, the third as the *constant rate model (vaccinated only)*, and the fourth as the *time-varying rate model (vaccinated only)*. In all four models, as result of behavioural relaxation, individuals in the NC compartments have a risk of infection $r$ ($r > 1$) times higher than that of compliant individuals [24,25,28,29].

It is important to note how the Health Belief Model provides the theoretical constructs to justify and interpret the mechanisms behind the potential transitions between compliance and non-compliance [53–55]. As mentioned, the adoption of health behaviours is affected,

**Table 1. Summary description of the four behavioural models.**

| Behavioural model | Key features |
| --- | --- |
| Constant rate model | Both susceptible ($S$) and susceptible vaccinated ($S^V$) individuals might become non-compliant. Transitions to (from) non-compliance take place at constant rate $\alpha$ ($\gamma$) |
| Time-varying rate model | Both susceptible ($S$) and susceptible vaccinated ($S^V$) individuals might become non-compliant. Transitions to non-compliance are function of the fraction of vaccinated individuals and $\alpha$. Transitions to compliance are function of the number of daily deaths per 100,000 and $\gamma$ |
| Constant rate model (vaccinated only) | Only susceptible vaccinated ($S^V$) individuals might become non-compliant. Transitions to (from) non-compliance take place at constant rate $\alpha$ ($\gamma$) |
| Time-varying rate model (vaccinated only) | Only susceptible vaccinated ($S^V$) individuals might become non-compliant. Transitions to non-compliance are function of the fraction of vaccinated individuals and $\alpha$. Transitions to compliance are function of the number of daily deaths per 100,000 and $\gamma$ |

among other factors, by perceived risks and susceptibility [53–55]. In this context, the start and progression of COVID-19 vaccinations might have changed the risk perception of some leading to behavioural relaxation [24,25]. On the other hand, a worsening of the epidemic conditions in one's social circle or in the total population might lead to another change in risk perception and/or susceptibility thus inducing individuals to resume their adoption of NPIs.

## Vaccines rollout, epidemic progression and NPIs in the four regions under study

The rollout of COVID-19 vaccines is a key part of our work. Hence, we start by providing some information about the initial phases of vaccinations in the four regions under study. In Fig 1A, we show the 7-day moving average of the fraction of daily newly vaccinated individuals across all age groups (shaded areas) and in the 70+ age group (solid lines) from the start of the vaccination rollout until the end date of the period under consideration. COVID-19 vaccination campaigns started on 2020/12/19 in British Columbia, 2020/12/08 in London, 2020/12/27 in Lombardy, and 2021/01/18 in São Paulo [34–38]. In all locations, we observe a peak in the first month during which the initial doses were mainly administered to front-line workers and fragile individuals. A similar behaviour can be seen for the 70+ age group. Moreover, the vaccine rates of this group show a second peak earlier with respect to the overall vaccination rates in the four regions, reflecting the priority given to the elderly population. Additionally, we observe that the vaccination rate in London was concentrated during the second to the fifth month since the rollout started. However, in British Columbia and Lombardy, vaccination started on a wider scale (i.e., beyond the prioritization of fragile individuals and frontline workers) from the third month of the rollout, and even later in São Paulo (from the fourth month).

To better understand the epidemic contexts in the periods under examination, next we discuss the evolution of the Pandemic and of NPIs in the four regions. In Fig 1B we show the confirmed deaths per 100,000 (solid lines). We can observe differences in the timing, shape, and intensity of the Pandemic across the four regions. Indeed, within the time horizon of interest, British Columbia and London experienced a single peak concentrated around the end of 2020. The peak in London was particularly intense and fuelled by the spread of the Alpha VOC which was initially detected there and traced back to a set of transmission chains that occurred in September 2020 [56]. However, in British Columbia, the peak of weekly deaths was more than five times lower compared to the other three regions and followed by a slower decrease with fluctuations due to the Alpha variant replacing the wild type [57]. In

Lombardy, we observe an intense peak (comparable to London) right before the end of the year and a second, less pronounced, peak in early April mainly due to the lifting of some of the NPIs and the spreading of Alpha [4]. Also in São Paulo we observe two intense peaks, which however are much closer both in terms of intensity and timing. Genomic data in this region suggests that the two peaks were driven by the rapid spread of the Gamma VOC first followed by the arrival and spread of the Delta VOC [43,58,59]. We note that the first of these two peaks take place in April, hence months later than the main peak in the other three areas here under examination.

In Fig 1B we also show the effect of NPIs on contacts estimated from the COVID-19 Community Mobility Report released by Google LLC [39] and the Oxford Coronavirus Government Response Tracker (OxCGRT) [40]. We use this data to estimate the contact levels during the Pandemic with respect to a pre-Pandemic baseline. Indeed, this data has been often used as a proxy of NPIs adoption, especially during the first two years of the Pandemic [17]. We refer the reader to the Materials and Methods for details. The plot suggests that, among the regions considered, individuals in London adopted the strictest NPIs. This is likely due to the emergence and rapid spread of the Alpha VOC that resulted in strong social distancing policies. These measures led to a significant reduction in contacts ranging from 10% to 50% with respect to the pre-pandemic contact levels. A similar trend, though not as strong, is observed in Lombardy where contacts rapidly dropped as the 2020 winter season progressed. In British Columbia, despite a visible drop at the end of 2020, we observe a level centred around 50% with respect to the pre-pandemic baseline. In São Paulo, we observe a much steeper increase from week 13 of 2021. By the end of the observed period, the contact level achieved over 80% with respect to the pre-pandemic baseline, indicating a significant rebound toward the pre-pandemic level.

## Baseline calibration

In Fig 1C, we show the weekly deaths as reported by official surveillance and as estimated by the baseline epidemic model in the four regions considered. In the figure, we plot medians along with 90% confidence intervals (CI) computed considering 1000 stochastic trajectories sampled from the joint posterior distributions estimated via the ABC-SMC calibration (see Materials and Methods for details). We highlight the start of the vaccination campaign in each location with vertical solid lines. To account for local differences in the epidemic trajectory, the starting point of our simulations is left as a free, calibrated, parameter (see Materials and Methods for more details). Furthermore, the simulation horizons in the four regions are set to capture the local epidemic wave(s) in the first months of the vaccine rollout. More precisely, we run simulations until 2021/07/04 in British Columbia, London, and Lombardy. In case of São Paulo we run until 2021/10/03 to capture the late waves of infection experienced there with respect to the other regions. Interestingly, most reported data points fall within the 90% CI of the calibrated baseline, which suggests the effectiveness of the model in fitting the unfolding of the Pandemic in the four locations despite the differences in terms of shapes and intensities of each epidemic curve. The analysis of the posterior distributions of free parameters, shown in S1 Text (Sect 5.4), indicates London as the region with the highest basic reproductive number $R_0$. This is likely due the dominance of the Alpha VOC in London at the start of our simulation window. Overall, the posterior estimates for $R_0$ are in line with previous findings [60–62]. More precisely, Ref. [60] reports values just above 1 and below 1.5 in British Columbia in October 2020. We found $R_0 = [1.1 – 1.5]$ (90% CI) at the start of our simulation period in the same month. Ref. [61] estimated values of $R_0 \sim 1.25$ in Lombardy at the end of October 2020. This finding is in line with our posterior estimates of $R_0 = [1.04 – 1.80]$ (90%

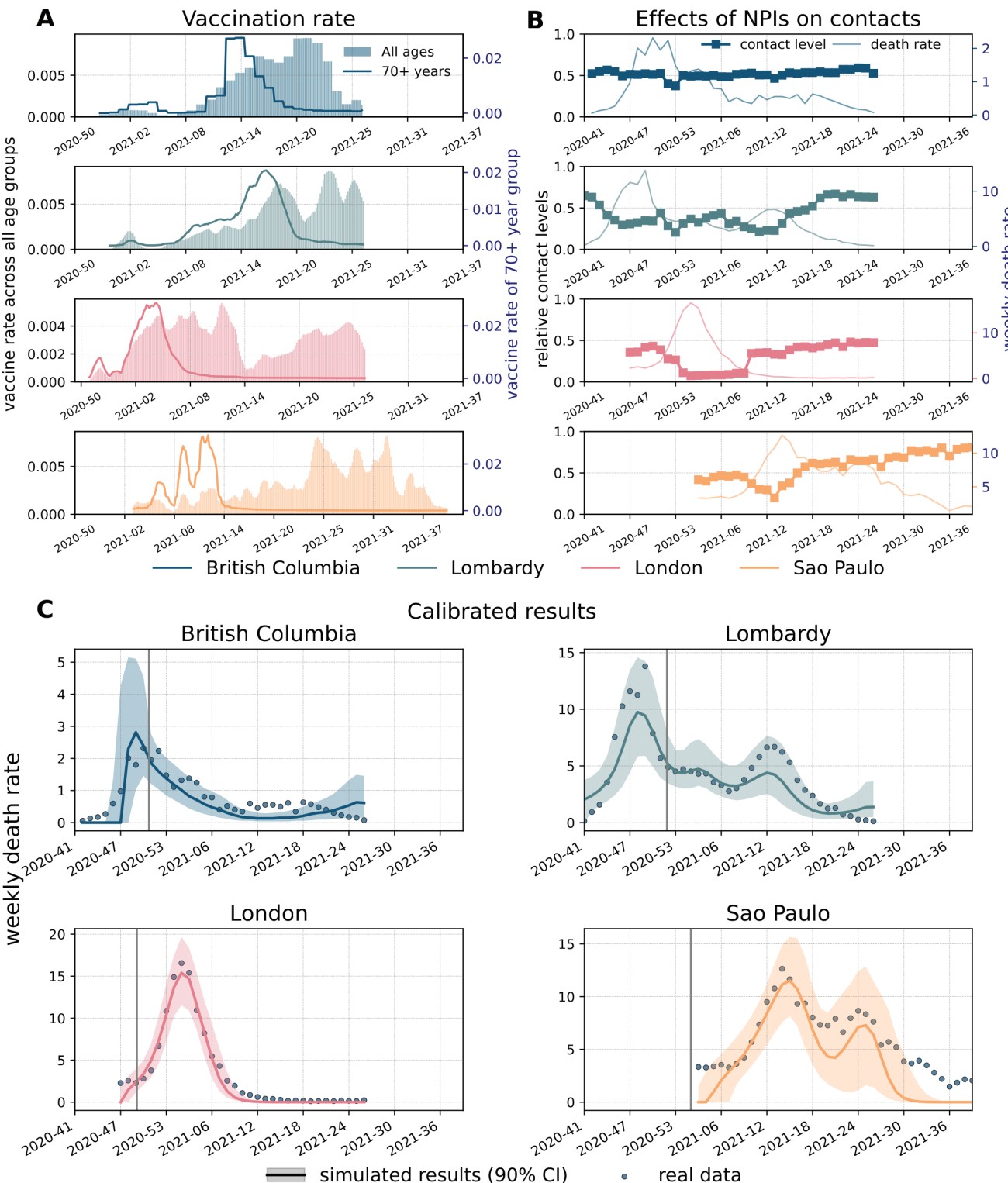

**Fig 1. Weekly deaths, vaccinations, contacts reduction, and calibration of baseline model.** The x-axis indicates the year and week. A) Fraction of daily newly vaccinated individuals across all age groups (shaded areas) and within the 70+ age-group (solid lines) in the four regions from up to down: British Columbia, Lombardy, London, and São Paulo. B) Contact levels during the Pandemic with respect to a pre-pandemic baseline. C) Reported data describing weekly deaths per 100,000 (dots) and the results from the calibrated baseline model (solid lines representing the medians, shaded areas the 90% confidence intervals). The grey vertical lines mark the start of vaccinations in different regions.

CI) in the same period. Ref. [62] reports values between 1.5 and 2 for Brazil in January 2021. Our estimates suggest $R_0 = [1.4 - 2.3]$ (90% CI) at the starting point of the simulation period in the same month. Despite these similarities it is important to note how the calibrated values of $R_0$ are affected by the model structure and specifically by how the force of infection is described. Thus, comparisons of fitted values across different models should be carefully interpreted [63].

## Estimating the impact of vaccines and NPIs via counterfactual scenarios

To estimate the impact of vaccines on COVID-19 deaths and infections we run a counterfactual scenario where they are removed from the baseline model. To this end, we first calibrate the model in the four locations. Then, we run matched simulations where we remove vaccinations. Hence, we quantify the effect of vaccines by computing the relative deaths difference (RDD) between the total number of deaths in a model with vaccines and those observed in an equivalent model without vaccines (see more details in Materials and Methods). The RDD for vaccines is shown in Fig 2A. The median values of RDD are greater than zero highlighting the clear positive effects of vaccinations. In particular, we find an RDD of 10.33% (90% CI: [−2.50%,25.01%]) in British Columbia, 15.90% ([9.65%,23.99%]) in Lombardy, 1.20% ([−0.06%,4.82%]) in London, and 50.69% ([45.77%,56.16%]) in São Paulo. The difference in RDD across the four regions is possibly due to several factors including timing and coverage of vaccines, local epidemiological context (e.g., VOC circulating), and NPIs in place. Notably, São Paulo, which exhibits the highest RDD, achieved also the highest vaccine coverage (78% of the population) by the end date of the simulation window. This is significantly larger than the coverage in British Columbia (69%), Lombardy (63%), and London (61%). As noted above, São Paulo is the region with the lowest reduction of contacts due to NPIs. This might contribute to enhance the role of vaccines. Notably, the median value of RDD in São Paulo corresponds to more than 70$K$ additional deaths averted (see S1 Text Sect 3.5). Though British Columbia has marginally higher vaccine coverage compared to Lombardy, its RDD is however the second lowest. This discrepancy can be explained by the relatively lower and slower epidemic progression in this region. As mentioned, this location experienced, at the peak, a burden of the disease about five times lower than the other three. The RDD values in this region are reflective of a very small absolute difference of deaths between the two scenarios (144 deaths as shown in S1 Text Sect 3.5). The RDD in London is the lowest value. In London, the early 2021 wave was fuelled by the Alpha VOC which was more transmissible and able to reduce the vaccines' protection from infection. Furthermore, the weekly deaths peaked when only a small fraction of the population was vaccinated and the vaccine coverage, by the end of our simulation window, is the smallest in the group of countries under investigation. Additionally, as shown above, London faced the strictest NPIs among the regions considered bringing contacts down even to 10% of pre-Pandemic levels.

Analogously, we compute relative difference of infections (RDI), defined as the fraction of total infections avoided by vaccines with respect to the infections observed in an equivalent model without vaccines (see S1 Text Sect 3.3 for details). In general, the RDIs are lower compared to RDDs across the regions as COVID-19 vaccines are more effective in preventing severe outcomes rather than infections. The RDIs show a similar pattern to that observed for RDDs across the four regions, except for São Paulo. The median of RDIs is only 9% while it is 51% for RDDs. This is likely due to São Paulo experiencing distinct viral strains, specifically the Gamma and Delta variants, which significantly reduced vaccine efficacy against infection compared to the other regions that instead saw the circulation of the wild type and the Alpha variant [44].

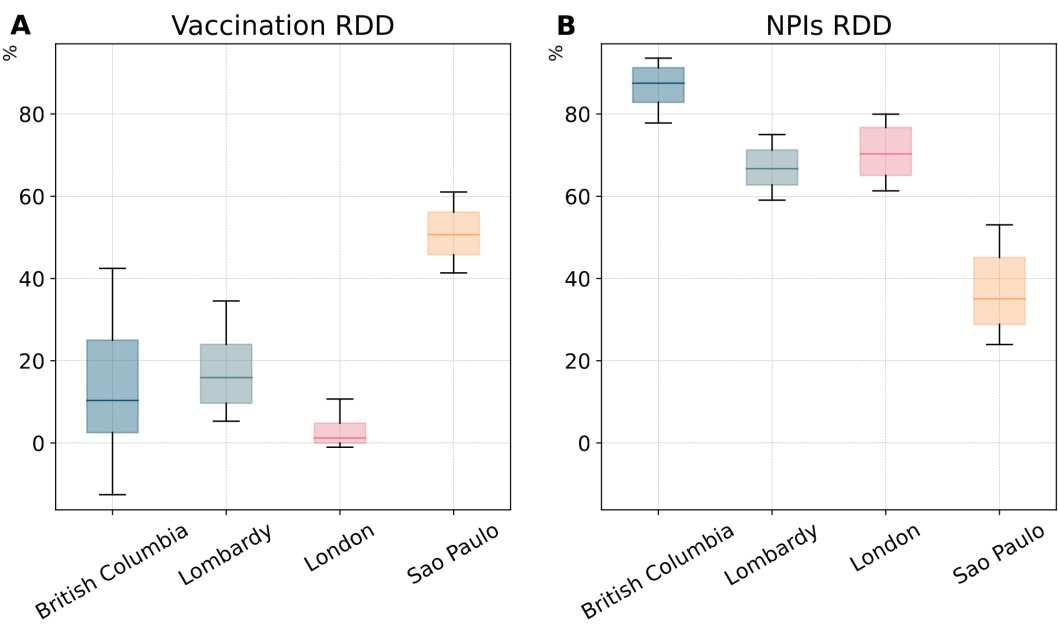

**Fig 2. Impact of vaccines and non-pharmaceutical interventions on COVID-19 deaths (baseline model).** A) Relative deaths difference (RDD) for vaccines. B) RDD for NPIs. The boxplots in both panels show the results considering 1000 stochastic trajectories in each region. The horizontal line within each box marks the median value, while the top and bottom edges correspond to the 90% CI. The whiskers extend to the maximum and minimum values. These estimates are obtained considering the baseline model.

We also investigate when the vaccination starts to have macroscopic effects by estimating the time when weekly deaths with and without vaccination differ by more than 1%. The results show that the difference of deaths began to diverge after 18 weeks since the vaccine rollout in British Columbia, 6 weeks in Lombardy, 8 weeks in both London and São Paulo (see S1 Text Sect 3.1).

To investigate the effects of NPIs on the progression of the Pandemic, we run a counterfactual scenario where we remove the impact of NPIs on contacts, while maintaining vaccinations. By comparing the trajectory of deaths with/without contacts modulation induced by NPIs, we find that without NPIs we would have experienced a larger number of deaths across all regions considered. Specifically, our results show that removing NPIs would have resulted in a much higher peak of weekly deaths during the period considered, 5.3 (90% CI: [3.3, 9.4]) times higher in British Columbia, 8.8 (90% CI: [6.8, 11.9]) times higher in Lombardy, 6.7 (90% CI: [5.5, 8.2]) times higher in London, and 4.7 (90% CI: [3.9, 5.7]) times higher in São Paulo compared to the estimates of the model considering NPIs. The absence of NPIs would have led to a 3 weeks earlier peak of weekly deaths in London and Lombardy, and 5 weeks São Paulo (see S1 Text Sect 3.2). Furthermore, we quantify the effect of NPIs by computing the fraction of total deaths avoided by NPIs with respect to the deaths observed in an equivalent simulations without NPIs (denoted by RDD as above). As shown in Fig 2B we find that 87.50% (90% CI:[82.82%,91.27%]) deaths have been avoided due to NPIs in British Columbia, 66.72% ([62.75%, 71.25%]) in Lombardy, 70.29% ([65.06%, 76.73%]) in London, and 35.07% ([28.82%, 45.12%]) in São Paulo. Not surprisingly, the RDD of the four regions are strongly correlated with their contact reduction with an exception in British Columbia. Specifically, British Columbia shows the highest RDD despite not featuring the strongest reduction in contacts. As noted above, the peak height of weekly deaths in British Columbia is more than five

times smaller than in the other regions. Given these conditions, removing NPIs brings the system into a different dynamical regime characterized, relatively speaking, by a much higher number of deaths. The RDD of London is the second largest due to the strict NPIs implemented during the observed period, imposed by the emergence and spreading of the Alpha VOC in September 2020. In contrast, São Paulo exhibits the lowest RDD, due to the relative low reduction in contacts induced by NPIs.

By comparing panels A and B in Fig 2, we see that, in the first months of the vaccine roll-out, NPIs averted more deaths than vaccinations. This result highlights the crucial role of NPIs in supporting the initial phases of vaccinations that, as discussed, struggled with significant challenges. Moreover, it shows the potential negative effects of behavioural relaxation in that period, given that the low vaccine uptake would be insufficient to offset a substantial decline in adherence to NPIs.

Analogously, we compute the fraction of total infections avoided by NPIs with respect to the infections estimated by an equivalent model without NPIs (denoted as RDI). The results of RDIs of the four regions are consistent with the results of RDD, except for London where we find slightly lower RDIs than Lombardy (see S1 Text Sect 3.3 for details).

## Estimating the extent of behavioural relaxation induced by vaccines

Building on the baseline and the literature, we developed four behavioural models where we incorporate behavioural relaxation mechanisms potentially induced by vaccines. In Fig 3, we show the calibrated results of all models (including the baseline) by presenting the medians and the 90% confidence intervals of weekly deaths. The calibrated curves are all consistent with reported epidemiological data. The differences between the models appear minimal to a visual inspection. To better investigate the nuances, we computed weighted mean absolute percentage errors (wMAPEs) as well as the Akaike Information Criterion (AIC) [64], and the Bayesian Information Criterion (BIC) [65]. The wMAPE measures the difference between the median outcomes of our models and reported data, while AIC and BIC scores assess the performance of models by trading off their complexity and the goodness of fit. Based on the AIC/BIC scores, we further calculate the AIC/BIC weight of each model for a more intuitive interpretation [65,66]. These weights can be interpreted as the probability that a model, among those considered, is the most likely given the empirical data [65,66]. While we only display AIC/BIC weights in the main text, we refer the reader to S1 Text (Sect 5.1) for the AIC/BIC scores.

The wMAPEs of the median of the five models are shown in Table 2. Behavioural models lead to smaller errors than the baseline in three regions out of four, nonetheless the best model results in limited improvements. Indeed we find a decrease in wMAPE of 9.8% in Lombardy, 2.0% in London, and 6.1% in São Paulo compared to the baseline. In more detail, the *constant rate model*, the *constant rate model (vaccinated only)*, and the *time-varying rate model* achieve, respectively, the lowest wMAPE in Lombardy, London, and São Paolo.

The picture changes, when we account for the complexity of the models. Indeed, according to AIC weights, the baseline is the most likely model in three regions (see Table 3). In Lombardy, instead, the *constant rate model* (followed by the baseline) emerges as the most likely. In the same table we can see that when considering BIC weights, the baseline is the most likely model across all regions. This metric is known to penalize more models with a larger number of free parameters, thus favouring simpler frameworks [67]. Nevertheless, in Lombardy the baseline is only marginally better than the *constant rate model*. These results show that, although behavioural mechanisms improve the goodness of fit in three regions, they come at the cost of an increased complexity that does not always offset the gains of fits. Furthermore,

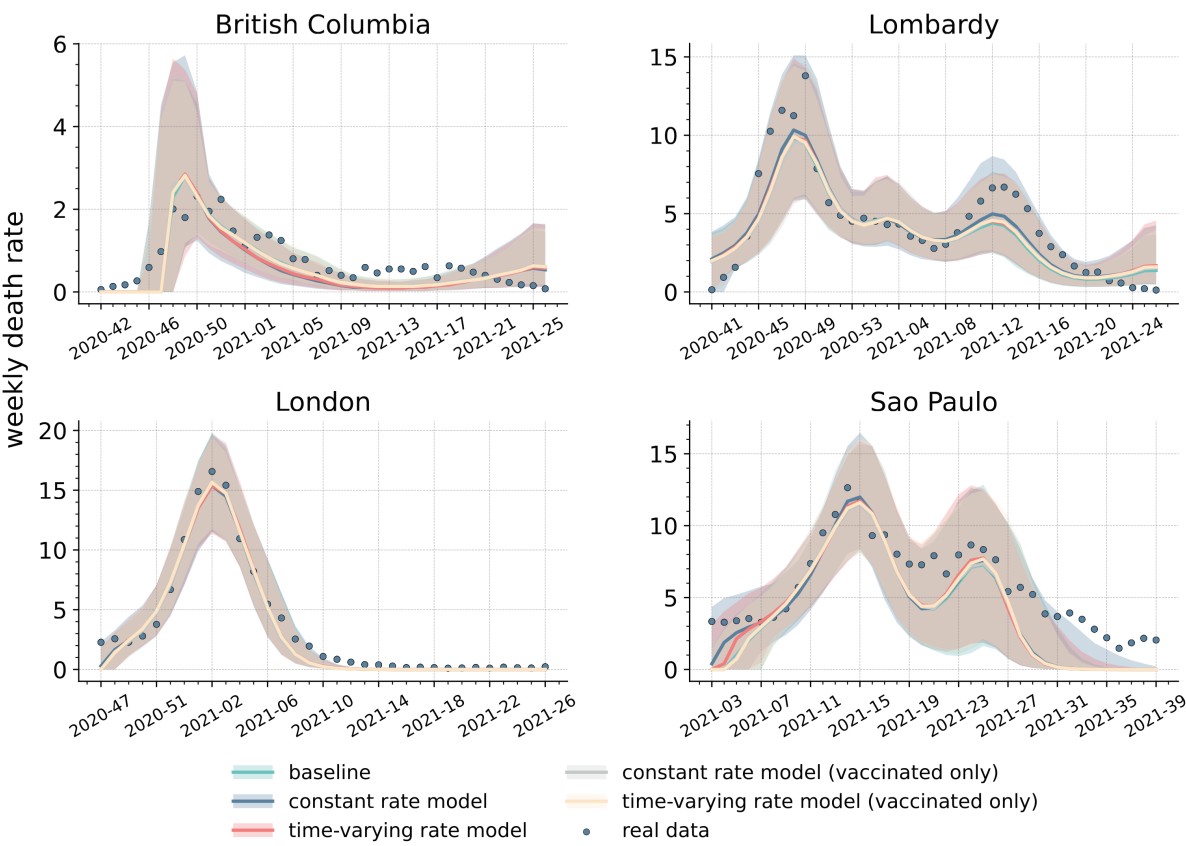

**Fig 3. Comparison of baseline and behavioural models.** Calibrated weekly deaths trajectories (i.e., weekly deaths per 100,000) for the baseline and four behavioural models across the four regions. Solid lines indicate the medians, while the shaded areas the 90% confidence intervals. Reported weekly deaths are denoted by blue dots.

**Table 2. wMAPEs obtained comparing the medians of calibrated models and reported weekly deaths. The lowest wMAPE in each location, indicating best performance, is highlighted in bold.**

| wMAMPE | Baseline | Constant rate | Time-varying rate | Constant rate (vaccinated only) | Time-varying rate (vaccinated only) |
|---|---|---|---|---|---|
| British Columbia | **0.3751** | 0.4431 | 0.4406 | 0.3862 | 0.3833 |
| Lombardy | 0.2742 | **0.2473** | 0.2702 | 0.2760 | 0.2686 |
| London | 0.1508 | 0.1529 | 0.1561 | **0.1479** | 0.1483 |
| São Paulo | 0.3056 | 0.2889 | **0.2869** | 0.3009 | 0.2959 |

due to the similarity of the outcomes, the selection of the most likely model is function of the metric considered (i.e., AIC vs BIC). Following Occam's razor principle, we can conclude that the inclusion of behavioural relaxation mechanisms is not fully justified, at least when looking at weekly deaths in the four regions studied. A simpler model, that does not explicitly account for this phenomenon, appears well suited to reproduce the unfolding of reported deaths. As shown in S1 Text (Sect 5.2), the ranking of the models in terms of Akaike weights does not change by removing the last 1,2,3,4 week(s). In the case of BIC weights, when removing the last 2,3,4 weeks we find the *constant rate model* to be more likely than the baseline, although the difference between the two remains small. Overall, the results are robust to the choice of

**Table 3. AIC and BIC weights computed considering calibrated models' medians and reported weekly deaths. The highest AIC/BIC weight in each location is highlighted in bold.**

| AIC | Baseline | Constant rate | Time-varying rate | Constant rate (vaccinated only) | Time-varying rate (vaccinated only) |
|---|---|---|---|---|---|
| British Columbia | **0.95** | 0.00 | 0.00 | 0.03 | 0.02 |
| Lombardy | 0.09 | **0.88** | 0.02 | 0.00 | 0.01 |
| London | **0.84** | 0.02 | 0.02 | 0.04 | 0.08 |
| São Paulo | **0.64** | 0.12 | 0.13 | 0.05 | 0.06 |
| **BIC** | | | | | |
| British Columbia | **0.99** | 0.00 | 0.00 | 0.00 | 0.00 |
| Lombardy | **0.54** | 0.44 | 0.01 | 0.00 | 0.01 |
| London | **0.98** | 0.00 | 0.00 | 0.01 | 0.01 |
| São Paulo | **0.95** | 0.02 | 0.02 | 0.01 | 0.01 |

**Table 4. Relative deaths difference (medians and 50% confidence intervals) for behavioural mechanisms. Numbers indicate percentages.**

| RDD (%) | Constant rate | Time-varying rate | Constant rate (vaccinated only) | Time-varying rate (vaccinated only) |
|---|---|---|---|---|
| British Columbia | -0.48 [-4.73,3.54] | -0.12 [-3.26,2.58] | 0.27 [-2.59,3.02] | 0.10 [-1.83,1.92] |
| Lombardy | -4.34 [-16.46,-0.79] | -0.58 [-1.67,0.22] | -0.33 [-1.08,0.31] | -0.14 [-0.77,0.39] |
| London | -1.91 [-8.57,-0.34] | -0.21 [-0.85,0.29] | -0.01 [-0.53,0.52] | 0.00 [-0.23,0.24] |
| São Paulo | -30.35 [-67.68,-7.91] | -5.18 [-25.42,-0.23] | -1.01 [-2.69,-0.2] | -0.45 [-1.5,0.03] |

the time horizon considered and are not affected by possible fluctuations in the tails of the epidemic curves.

As shown in S1 Text (Sect 5.4), the posterior distributions of increased transmissibility (i.e., $r$) of non-compliant individuals range between $10 - 30\%$, which is a significant increase. However, with the exception of São Paulo in case of the *constant rate model*, the median fraction of non-compliant individuals at each time step is smaller than 20% (see S1 Text Sect 4.2). In other words, the calibration selects regions of the phase space where the population of non-compliant individuals remains a minority. This result corroborates the lack of clear signs of behavioural relaxation on weekly deaths. Indeed, even assuming the presence of such phenomenon, the empirical evidence constraints it to a small group of the total population.

In order to gather a better understanding of the dynamics at play, and to isolate the potential effects of behavioural relaxation on deaths, we run another counterfactual analysis removing the relaxation mechanisms in the four calibrated behavioural models. In doing so, we compute the relative deaths difference (RDD) between the models with/without behavioural relaxation. As shown in Fig 4 and in Table 4, the median RDD values are below zero in the large majority of cases, indicating that removing behavioural relaxation generally results in fewer deaths. The results show that the *constant rate model* and the *time-varying rate model* lead to a larger difference in deaths (especially the *constant rate model* in São Paulo) compared to the versions of the models that restrict behavioural relaxation to vaccinated individuals. Not surprisingly, the impact of non-compliance extended to the whole population is higher. It is important to notice that, with two exceptions (i.e., the *constant rate model* and the *time-varying rate model* in São Paulo), the RDD values are close to zero. As noted above, the posterior distributions of behavioural parameters selected in the calibration lead to configurations where relaxation does not strongly impact deaths.

Finally, we compute the relative infection difference (RDI) which shows consistent results with those of RDDs (see S1 Text Sect 3.4).

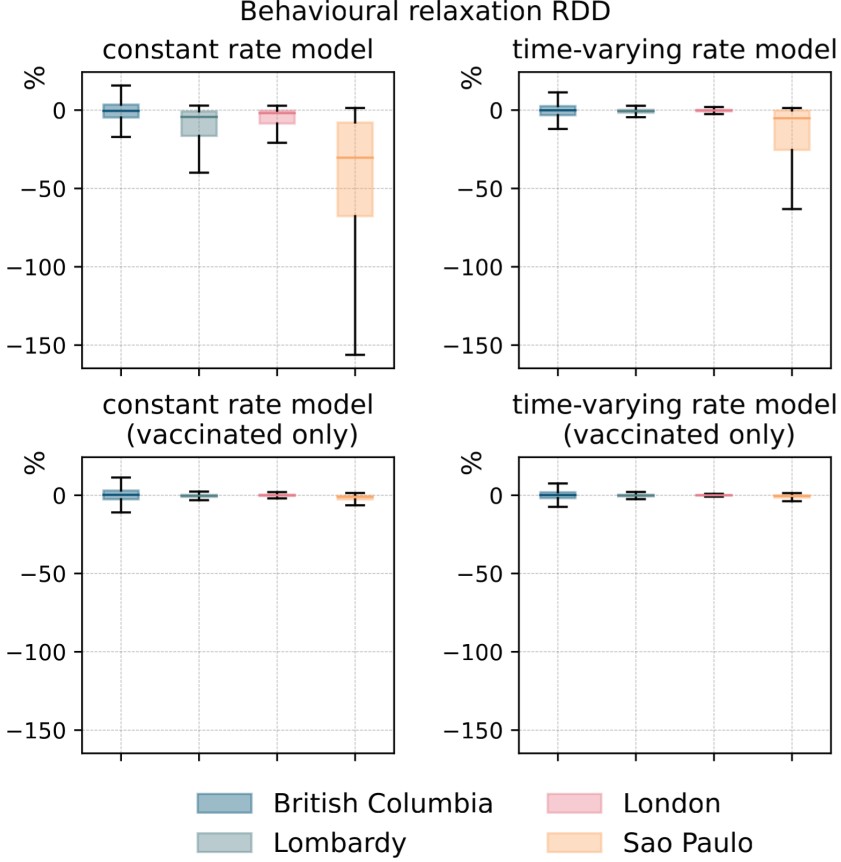

**Fig 4. The impact of behavioural relaxation on COVID-19 deaths (behavioural models).** We plot the RDD (i.e., relative deaths difference) for behavioural mechanisms in the four models and regions. Each boxplot is built considering 1000 stochastic simulations. The horizontal line within each box marks the median value, while the top and bottom edges correspond to the 50% CI. The whiskers extend to the maximum and minimum values after removing the outliers that beyond 1.5 times the interquartile range.

## Discussion

In this paper, we aimed to find signs and to quantify the extent of behavioural relaxation possibly induced by vaccines during the initial phase of the COVID-19 vaccine rollout. To this end, we developed a series of stochastic epidemic compartmental models integrating age-structure, vaccinations, NPIs, variants of concern, and deaths. We used a baseline model, without any behavioural relaxation mechanism, as a reference, and on top of this, we developed four behavioural models that extended previous work to account for individual behaviours in response to vaccination and the epidemic [22–27]. We tested these models considering weekly deaths in four regions: British Columbia (Canada), Lombardy (Italy), London (United Kingdom), and São Paulo (Brazil). These locations sample different epidemic, socioeconomic and socio-demographic contexts, as well as different vaccine rollout schedules and coverages. We first calibrated the baseline model to reported data and studied two counterfactual scenarios to quantify the impact of vaccines and NPIs on COVID-19 deaths and infections. Our results confirmed that both significantly reduced mortality and infections. Furthermore, they highlighted the critical role of NPIs in supporting the challenging initial phases of

vaccinations. We then calibrated the four behavioural models and compared them considering both their goodness of fit and complexity. Behavioural models estimates are closer to real data than the baseline in three locations, though the improvements are limited between 2% and 10% in terms of wMAPE. Behavioural mechanisms increase models' complexity which is not always offset by the benefits of improved fits. This suggests that additional mechanisms of behavioural relaxation linked to vaccination may not be evident across all regions. Furthermore, our results suggest that, even if behavioural relaxation took place, it was limited to a minority of the population.

Overall, our results indicate that behavioural relaxation did not leave clear marks on reported deaths. This finding, in line with surveys conducted in France [32], might be interpreted as a lack of support for systematic behavioural relaxation induced by COVID-19 vaccines. However, our findings do not exclude that, in fact, behavioural relaxation took place as suggested by other surveys [28–31]. As mentioned above, assuming the presence of behavioural relaxation, the calibration with weekly deaths constraints the phase space to regions where the fraction of non-compliant individuals is a minority. This might explain the good performance of the baseline: the impact of behavioural relaxation could be accounted for by simpler models that do not explicitly consider additional mechanisms. In our settings, the effects of behavioural relaxation might be fully captured by the modulation of contacts induced by NPIs.

It is important to mention that the selected target variable (i.e., weekly deaths) might have influenced our findings. Indeed, behavioural relaxation might have been more prevalent but not to the levels needed to affect mortality levels observed at a macroscopic scale. Signs of behavioural relaxation might be clearer in other indicators. For example, given the strong dependence of COVID-19 mortality on age [68], behavioural relaxation might have primarily affected infections, especially among the young, active population, rather than deaths. However, data on confirmed cases has been shown to be a poor indicator and a very hard signal to fit due to under-reporting and variations in testing policies among other factors [69].

These possible interpretations of our results highlight the issue of non-identifiability of complex behavioural mechanisms in epidemic models linked to highly degenerate phase-spaces and to the interplay among the various processes at hand (e.g., disease transmission and behavioural reactions) [70,71]. Arguably, the quest towards a clear identification of behavioural reactions to epidemics is linked to the use of multi-stage calibration steps informed by a range of data types and indicators that go beyond the solely use of epidemic variables [70]. For example, future work is needed to explore whether the integration of behavioural data, such as aggregated mobility metrics, into the calibration pipeline might provide further constraints and help reduce the problem of non-identifiability. Progresses in this direction are contingent to advances in data collection and data sharing as well as to the identification of key behavioural observables and novel data streams to track [72,73].

Our work comes with limitations. First, the epidemiological and vaccination data are sourced from different datasets. Although the data has been obtained from official sources, the granularity provided is not homogeneous. Second, we considered a simplified vaccination protocol assuming that only susceptible individuals can get vaccinated and a single dose regiment. These assumptions have been made to simplify the model structure. Due to the lack of data for all VOCs studied, we did not account for the effects of vaccines on the infectiousness of breakthrough cases (i.e., the infectiousness of vaccinated individuals that acquire infection) [74]. Nevertheless, in the four regions under study here, the larger number of infections took place when the fraction of vaccinated individuals was still small. Hence, we do not expect this simplification to significantly affect the results. Third, we used regional-level

data regarding school closures for British Columbia, London, and São Paulo but country-level data for Lombardy, due to the lack of specific regional data within Italy. Fourth, there is lack of available data to parametrize the rates regulating behavioural relaxation. As a consequence, we had to calibrate the behavioural parameters within some rather arbitrary ranges. Fifth, we did not consider that behavioural relaxation, nor the increase in infection risk of non-compliant individuals, might be age dependent. Indeed, given the lack of detailed data for all age-groups, introducing age-dependent parameters would have drastically increased the complexity of the models exacerbating the issues of non-identifiability. Furthermore, as a way to simplify the structure of the models, we assumed that only susceptibles individuals might relax their behaviours. In doing so, we excluded the possibility that also infectious individuals might become non-compliant. Arguably, the relatively short infectious period (i.e., 2.5 days) limits the impact of this choice on the findings. Sixth, we modulated contacts rates using mobility data towards workplaces and other public locations. We did not consider changes in contacts rates at home. Assuming an homogenous mixing, variations in contact rates scale quadratically with respect to changes in mobility (i.e., visits to particular locations). However, as the risk of infection does not increase quadratically if two or more people remain confined at home, it is unclear to what extent this relationship holds in households. Seventh, though we accounted for higher transmission rates, shorter latent periods, and decreased vaccine efficacy against VOCs, we used the same infection fatality rate (IFR) across all strains and regions. Eighth, the results reported above are obtained using a relatively simple compartmental model that neglects both presymptomatic and asymptomatic stages of the infection. As sensitivity analysis, in S1 Text, we report the results obtained modifying the model structure to include these aspects. The overall picture remains unchanged. Finally, our model does not account for socioeconomic nor socio-demographic differences in vaccines uptake nor in adoption of NPIs [75,76].

Overall, our work highlights the critical importance of both NPIs and vaccines in curbing COVID-19 deaths and infections during the initial months of COVID-19 vaccination campaigns. Our findings pave the way for further research to refine the proposed models and deepen our understanding of the interaction between individual protective behaviour and vaccinations in a broader context.

## Materials and methods

### Baseline model

As baseline we adopt a stochastic age-stratified epidemic compartmental model that integrates vaccination, NPIs, and the emergence/spread of multiple strains. The baseline is based on a Susceptible-Latent-Infected-Recovered (SLIR) compartmentalization extended to account for deaths. Individuals are grouped into 16 age brackets with a five-year interval (except for the last group which is 75+). We use age-stratified infection fatality rates (IFR) from Ref. [68], age-stratified contact matrices from Ref. [77], and official demographic data [78–81]. The natural history of the disease is modelled as follows. By interacting with the Infected ($I$), Susceptible ($S$) individuals transition to the latent stage ($L$ compartment) where they are infected but not yet infectious. We assume a force of infection $\lambda_k$ (i.e., the per capita rate at which susceptibles get infected) function of age (i.e., $k$), the transmissibility of each strain, contact matrices, and NPIs (see S1 Text Sect 1.1 for more details). Individuals stay in $L$ for an average of $\epsilon^{-1}$ days. After, they become infectious transitioning to the $I$ compartment. After the infectious period $\mu^{-1}$, infected individuals either recover with probability $(1 - IFR_k)$

(transitioning to the $R$ compartment) or die from the disease with probability $IFR_k$ (transitioning to the $D$ compartment). We also consider a delay of $\Delta$ days in deaths reporting. Therefore, individuals are moved to the $D^o$ compartment from $D$ after $\Delta$ days. The model does not explicitly consider isolation/quarantine of infectious individuals. However, we opted for a short infectious period assuming that infected individuals develop symptoms and isolate, or get tested and isolate on average after $\mu^{-1} = 2.5$ days.

**Modelling vaccinations.** We incorporate vaccinations into our models by doubling all the compartments to include vaccinated individuals at any stage of the disease. We assume that only susceptible individuals can receive the vaccine. Additionally, to simplify the model, we disregard the time interval between the first and second dose. Consequently, we assume that individuals acquire full protection right after the first inoculation. We use real data to capture the unfolding of the vaccination campaigns in the four regions [34–38]. Vaccinations are simulated as follows. For each age-group $k$, a fraction of susceptible individuals (non-vaccinated) transition to the compartment of susceptible vaccinated according to the number of daily vaccinations from data [34–38]. We note how vaccine data are reported weekly for British Columbia while daily for the others. Thus, in British Columbia, we convert the weekly number of doses into daily by splitting them homogeneously in each week day.

Vaccines protect individuals in two ways: by lowering the risk of infection, and by reducing the risk of death in case of breakthrough infection. In practice, for vaccinated individuals the force of infection is multiplied by a factor $1 - VE_S$, where $VE_S$ denotes the vaccine's efficacy against infection. If a vaccinated individual becomes infected, the IFR is further reduced by a factor $1 - VE_M$, where $VE_M$ represents the vaccine's efficacy against death. Therefore, the overall vaccine efficacy for a susceptible individual, against death, is $VE = 1 - (1 - VE_S)(1 - VE_M)$.

**Modelling the impact of NPIs on contacts.** We use Ref [77] to obtain pre-Pandemic contact matrices, i.e., **C**. These stratify contacts by age and location (i.e., context) such that:

$$\mathbf{C} = \mathbf{C}_{home} + \mathbf{C}_{work} + \mathbf{C}_{school} + \mathbf{C}_{others} \tag{1}$$

Hence, each element $C_{ij}$ describes the mean daily number of contacts of an individual in age group $i$ with individuals in age group $j$ across all contexts. We estimate weekly variations in these contact rates due to NPIs using mobility data from the COVID-19 Community Mobility Report released by Google LLC [39] and the Oxford Coronavirus Government Response Tracker (OxCGRT) [40]. In particular, we adjust contact matrices, at time $t$, as follows:

$$\mathbf{C}'(t) = \mathbf{C}_{home} + w_{work}(t)\mathbf{C}_{work} + w_{school}(t)\mathbf{C}_{school} + w_{others}(t)\mathbf{C}_{others}. \tag{2}$$

$\mathbf{C}'(t)$ is the adjusted contact matrix at time $t$. The values $w_{work}(t)$ and $w_{others}(t)$ are computed as $(1 - x(t)/100)^2$ where $x(t)$ is the percentage change in mobility to specific locations with respect to a pre-Pandemic baseline [39]. The baseline is defined by Google as the median value of mobility between January $3^{rd}$ and February $6^{th}$, 2020. The values of $w_{work}(t)$ are computed considering the specific field related to workplaces, while $w_{others}(t)$ considering an average of the fields related to retail, recreation and transit stations. We use a square form as the number of contacts in a location scales proportional to the square of people visiting that location [4,5]. Furthermore, $w_{school}(t)$ is computed as $(3 - school(t))/3$, where $school(t)$ is an index measuring the strictness of containment policies in schools reported by OxCGRT at time $t$ [40]. It takes integer values from 0 (i.e., no containment measures are in place) to 3 (i.e, full school closure). Although we acknowledge that contacts at home may have increased due to the adoption of NPIs, we opted for not modifying them as function of time. Indeed,

due to saturation effects, we expect the link between changes in the time spent at home and infection risk to be more complex than for other locations. In S1 Text we show that the variation in contact patterns estimated as outlined above is consistent with the independent findings obtained in Ref. [82] via surveys. We note how the contact levels, with respect to the pre-Pandemic baseline, shown in Fig 1B are computed as $\left( w_{work}(t) + w_{school}(t) + w_{others}(t) \right)/3$.

**Modelling multiple viral strains.** In British Columbia, Lombardy, and São Paulo, we introduce new compartments $L'$, $I'$, $R'$, $D'$ and $D^{o'}$ to account for non-vaccinated individuals infected by a VOC (i.e., a second strain). Similarly, we consider compartments $L^{V'}$, $I^{V'}$, $R^{V'}$, $D^{V'}$, $D^{oV'}$ for vaccinated individuals infected by a VOC. We model the introduction of a second strain as follows. We denote $t_{var}$ as the time at which the second variant establishes its presence in each location. To this end, we use genomics data from Ref. [41] and calibrate $t_{var}$ in a range between 0 and 42 days prior to the first date in which each variant is consistently featured in the genomics data (i.e., the share of samples attributed to the variant in the genomics data is greater zero from this first date onwards). We did not consider the first appearance in the dataset as a single sample might be linked to an isolated importation from a different location. Then, at time $t_{var}$ we initialize the compartment $I'$ with a number of individuals equivalent to the 1% of people infected by the initial strain (i.e., $I$ and $I^V$).

The four regions studied faced the Alpha, Gamma, and Delta VOC. In detail, in British Columbia and Lombardy, Alpha appeared as a second strain replacing the wild type. In London, the Alpha variant was the dominant variant circulating throughout our time horizon. In São Paulo, the initial variant observed at the start of the simulation period was Gamma which was then replaced by Delta.

We model VOCs by adjusting (or not) relevant parameters. According to literature, the latent period of Alpha and Gamma is similar to that of the wild type [48–51]. Hence, we kept $\epsilon' = 3.7^{-1} days^{-1}$ for these variants. In contrast, the latent period of Delta has been reported to be shorter [52]. Hence, we set $\epsilon' = 3^{-1} days^{-1}$ for this VOC. For all variants, including the wild type, we set the infectious period to $\mu = 2.5^{-1} days^{-1}$. Additionally, the second variant may exhibit higher transmissibility. Thus we adjust the transmissibility of variants by multiplying it by a parameter $\sigma$, which represents the relative increase in transmissibility. Following the literature we set $\sigma = 1.5$ for Alpha compared to the wild type [44]. In São Paulo, where Delta replaced Gamma, we calibrate $\sigma$ in a range of [1.6–2.5], as no specific indication was found in the literature. Moreover, the vaccine efficacy might also be lower against variants [44,46,47]. Following the literature, we set the vaccine efficacy as $VE = 90\%$ ($VE_S$=85%) against the wild type [46], $VE = 85\%$ ($VE_S = 75\%$) against Alpha [46], $VE = 80\%$ ($VE_S = 65\%$) against Gamma [47], and $VE = 90\%$ ($VE_S = 60\%$) against Delta [46].

## Behavioural models

Building on the baseline and the literature, we implemented four additional models that also include behavioural relaxation mechanisms [28,30]. To this end, we introduce new compartments $S_{NC}^V$ and $S_{NC}$ to account for susceptible individuals that relax adoption of NPIs becoming non-compliant (NC). In detail, individuals who relax their behaviour transit from $S$ ($S^V$) to $S_{NC}$ ($S_{NC}^V$). Conversely, non-compliant individuals who return to compliant behaviours transition from $S_{NC}$ ($S_{NC}^V$) to $S$ ($S^V$). We assume that individuals in the non-compliant compartments get infected at higher rates compared to those in compliant compartments [24,25]. This is accounted for by multiplying their force of infection by a factor $r > 1$.

Following the literature, we investigate four behavioural models that differ for two key aspects: 1) the groups that might relax their behaviour and 2) the mechanisms used to describe how individuals enter and leave the non-compliant compartments. In the *constant*

*rate model*, susceptible individuals (vaccinated or not) can enter (leave) the NC compartments at constant rate $\alpha$ ($\gamma$). In the *time-varying rate model*, susceptible individuals (vaccinated or not) enter or leave the NC compartments at time-varying rates. The transition rate from $S$ ($S^V$) to $S_{NC}$ ($S_{NC}^V$) is set as a function of the fraction of vaccinated individuals and a parameter $\alpha$. The transition rate from $S_{NC}$ ($S_{NC}^V$) to $S$ ($S^V$) is set as a function of the number of reported daily deaths per 100,000 and a parameter $\gamma$. The *constant rate model (vaccinated only)* and the *time-varying rate model (vaccinated only)* are analogous to the previous two. However, in these only vaccinated individuals can transition to the NC compartment. The structure of our models is illustrated in Fig 5. More details are reported in the Supporting Information.

## Models calibration

We apply an Approximate Bayesian Computation-Sequential Monte Carlo (ABC-SMC) method to calibrate our models [33] to reported data. The goal of ABC-SMC algorithm is to estimate the posterior distribution of free parameters $\theta$ starting from an input prior distribution $P(\theta)$. It is an extension of the ABC rejection algorithm, where suitable parameters are found by iteratively sampling from the prior distribution and computing for each sampled parameter set $\theta_i$ a distance function $d(\mathbf{y}_i, \mathbf{y}_{data})$, where $\mathbf{y}_i \sim f(\theta_i)$ is the output of the

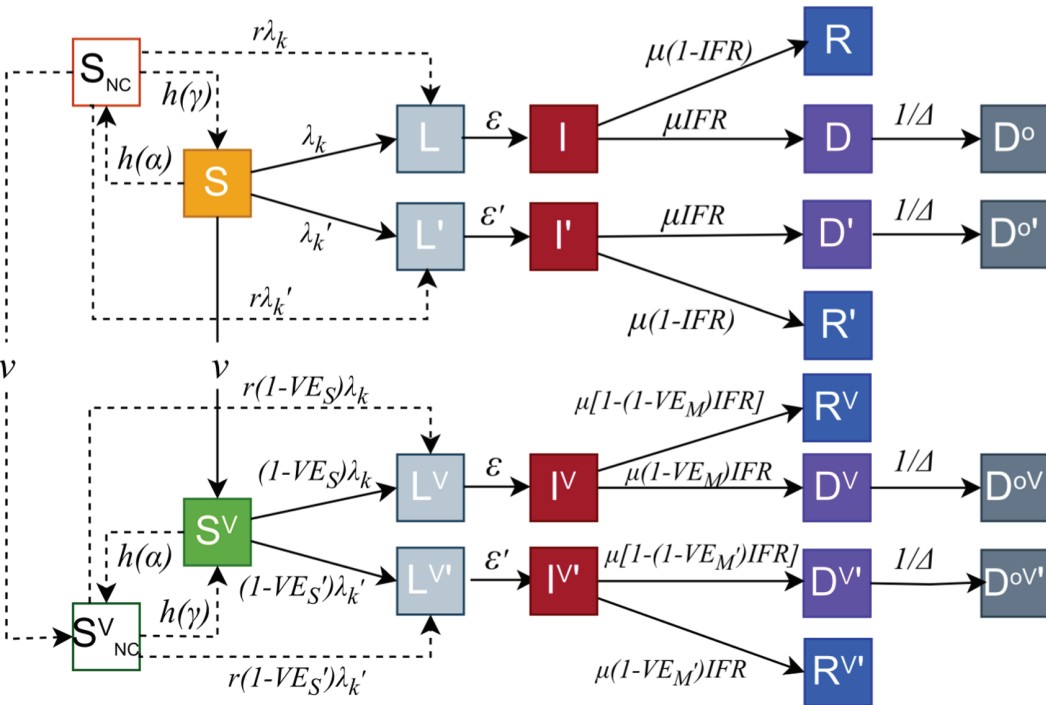

**Fig 5. Epidemic compartment structure.** All compartments connected by solid lines constitute the baseline model. This model includes susceptible ($S$), latent ($L$), infected ($I$), recovered ($R$), and dead ($D$, $D^o$) compartments. The top row represents non-vaccinated compartments, whereas the bottom row represents vaccinated compartments. Individuals in the $S$ compartment get vaccinated according to real vaccine rates ($\nu$) and then transition to the $S^V$ compartment. To account for the emergence of a second variant, we double the compartments creating $L'$, $I'$, $R'$, $D'$, and $D^{o'}$. This is done also for the vaccinated compartments that become $L^{V'}$, $I^{V'}$, $R^{V'}$, $D^{V'}$, $D^{Vo'}$. Behavioural models include susceptible non-compliant compartments ($S_{NC}$, $S_{NC}^V$) connected by dotted lines, where individuals have $r$ times higher probability of getting infected with respect to susceptible compliant individuals ($S$ and $S^V$). In the *constant rate model* and the *time-varying rate model*, we include $S_{NC}$ and $S_{NC}^V$, whereas in the vaccinated-only versions, we include only $S_{NC}^V$

model and $\mathbf{y}_{data}$ is the reported data (i.e., weekly deaths). Each $\theta_i$ is accepted if $d(\mathbf{y}_i, \mathbf{y}_{data}) \leq \xi$, where $\xi$ is a predefined tolerance. The process is repeated until $M$ parameters $\theta_i$ are accepted. Their distribution approximates the true posterior distribution $\Pi(\theta|\mathbf{y}_{data}, \xi)$. The ABC rejection algorithm is of straightforward implementation, however suffers from several limitations. First, the values of $M$ and of the tolerance $\xi$ are free parameters that shape the interplay between convergence speed and accuracy [83]. Second, the prior distribution is never updated to account for information from previous iterations. The ABC-SMC framework has been developed to tackle these issues. It consists of $T$ generations (i.e., iterations). The first one is based on a rejection algorithm step where $\xi$ is set to a high value. In the second generation, the tolerance is decreased, parameters are sampled from those accepted in the previous step and perturbed via a kernel to avoid converging on local minima of the phase space. The process is repeated for $T$ generations of $M$ particles (i.e, samples) each. Then, the set of accepted parameters in the last generation is used as the empirical posterior distribution. We adopted a Python implementation of ABC-SMC from the package *pyabc* [84].

The free parameters and the priors explored in our models are:

- Reproductive number $R_0$. We explore values in the interval [1,3].
- Delay in reporting deaths $\Delta$. Consistent with observations, we explore the interval [3,64] [85].
- Initial fraction of infections of the total population $i_{ini}$. We estimating the ranges from the number of deaths and IFR across the four regions. We explore the interval [0.0005,0.02].
- Initial fraction of individuals with residual immunity from past waves $r_{ini}$. We explore the range of [0.1,0.4] [86].
- Start date $t_0$ of the simulation of epidemic. We calibrate $t_0$ within a range of 8 weeks such that $t_0 = t^* - \Delta t$, where $\Delta t = [0, 1, \ldots, 7]$ week(s). The baseline dates $t^*$ are set as 8 weeks before the peak of mortality in real data. Following this, $t^*$ is set as 2020/10/12 for British Columbia, 2020/10/05 for Lombardy, 2020/11/16 for London, and 2021/01/18 for São Paulo.
- Introduction date of a VOC $t_{var}$ (applied for all regions except London). We use genomics data from Ref. [41] and calibrate $t_{var}$ in a range between 0 and 42 days prior to the first date $t^*_{var}$ from which each variant is consistently featured in the genomics data. Thus, $t_{var} = t^*_{var} - \Delta t_{var}$, where $\Delta t_{var} = [0, 1, \ldots, 42]$ day(s). $t^*_{var}$ is set as 2020/12/21 in British Columbia, 2020/9/28 in Lombardy, and 2021/3/29 in São Paulo.
- Relative transmissibility $\sigma$ of Delta with respect to Gamma. In the case of São Paulo, Delta replaced Gamma. Literature shows that Delta is about 1.3 times more transmissible than Gamma [43]. We explore the interval [1.0–2.5].
- Behavioural parameters $\alpha$ and $\gamma$. We explore the interval [0.0001–10]. For both and sample them on a logarithmic scale.
- Relative infection probability of non-compliant individuals $r$. Individuals who relax their behaviour are more likely infected. Therefore, we increase the infection probability of non-compliant individuals by multiplying it by a factor $r$. We explore the interval [1.0,1.5].

The initial prior distribution $P(\theta)$ is obtained sampling each interval uniformly.

## Model initialization

We initialize the number of individuals in each compartment as follows. We assume that, at the beginning, all individuals are in the compartments $S$, $L$, $I$, or $R$. The initial individual

numbers of infected (including both $L$ and $I$ compartments) and recovered ($R$) individuals are set as fractions of total population considering under-reporting and official data. The total number of infected individuals is then distributed to $L$ and $I$ compartments proportionally to the inverse of their respective transition rates. Besides, since our model is age-stratified, the initial numbers of individuals in compartments $S$ ($L$, $I$, or $R$) in each age group is set as $N_S \times N_k/N$ ($N_L \times N_k/N$, $N_I \times N_k/N$, or $N_R \times N_k/N$), where $N_k$ is the individual number in age-group $k$, $N$ is the total individual number, $N_S$ is the total number of individuals in compartment $S$. All parameters of our models are displayed in Table 5.

## Models evaluation

For evaluating the models we use weighted mean absolute percentage errors (wMAPEs), Akaike Information Criterion (AIC) scores [64], and Bayesian Information Criterion (BIC) scores [65]. wMAPE measures the difference between the median outcomes of our models and reported data. It is defined as:

$$\text{wMAPE} = \frac{\sum_{t=1}^{t_f} |y_{data,t} - median(y_{i,t})|}{\sum_{i=t}^{t_f} y_{data,t}} \tag{3}$$

where $y_{data,t}$ is the reported data at time $t$, $median(y_{i,t})$ is median trajectory of model $i$ at time $t$, and $t_f$ is the total number of weeks.

**Table 5. Models' parameters.**

| Parameter | Symbol | Value |
|---|---|---|
| Reproductive number | $R_0$ | Calibrated within [1,3]. |
| Latent period | $\epsilon$ | $3^{-1} days^{-1}$ for Delta [52], $3.7^{-1} days^{-1}$ for others [48–51] |
| Infectious period | $\mu$ | $2.5^{-1} days^{-1}$ [48,87] |
| Transmission rate | $\beta$ | Obtained from $R_0$ |
| Infection fatality rate | IFR | Ref. [68] |
| Contact matrix | **C** | Ref. [77] |
| Delayed days in reporting deaths | $\Delta$ | Calibrated within [3,64] [85] |
| Initial fraction of infections | $i_{ini}$ | Calibrated within [0.0005,0.02] |
| Initial fraction of recoveries | $r_{ini}$ | Calibrated within [0.1,0.4] |
| Adjustment of the start date of the simulation of epidemic with respect to the baseline date ($t^*$) | $\Delta t$ | Calibrated within [0,7] weeks |
| Adjustment of the introduction date of a VOC with respect to the date ($t_{var}^*$) at which each variant is consistently featured in the genomics data | $\Delta t_{var}$ | Calibrated within [0,42] days [88] |
| Relative transmissibility of a second variant | $\sigma$ | 1.5 for Alpha [44]; calibrated within [1.0–2.5] for Delta with respect to Gamma [43] |
| Overall vaccine efficacy | VE | 0.9 against wild type; 0.85 against Alpha variant; 0.8 against Gamma variant; 0.9 against Delta variant [46,47] |
| Vaccine efficacy against infection | $VE_S$ | 0.85 against wild type; 0.75 against Alpha variant; 0.65 against Gamma variant; 0.6 against Delta variant [46,47] |
| Transition rate towards non-compliance | $\alpha$ | Calibrated within [0.0001–10] |
| Transition rate towards compliance | $\gamma$ | Calibrated within [0.0001–10] |
| Relative infection risk of non-compliant individuals | $r$ | Calibrated within [1.0–1.5] |

AIC scores assess the performance by trading off the complexity and fitting of the models. The AIC score of model $i$ is computed as:

$$AIC_i = t_f log\Phi^2 + 2K_i \tag{4}$$

where $\Phi^2$ is the sum of the squares of residuals, $t_f$ is the number of data points (i.e., weeks considered), and $K_i$ is the number of free parameters of model $i$. To obtain a more intuitive metric, we calculate Akaike weights from the AIC scores. These can be interpreted as the relative likelihood of a given model [66]. The Akaike weight of model $i$, denoted by $w_i$, is computed as

$$w_i = \frac{e^{-\Delta AIC_i/2}}{\sum_{i=1}^{Q} e^{-\Delta AIC_i/2}} \tag{5}$$

where $\Delta AIC_i$ is the difference between the AIC score of model $i$ and of the best model (i.e., the one with lowest AIC score), and $Q$ is the number of models.

BIC scores are similar to AIC scores, but contain a different term to account for model's complexity. The BIC score of model $i$ is computed as:

$$BIC_i = t_f log\Phi^2 + log(t_f)K_i \tag{6}$$

To obtain a more interpretable comparison of models, we also calculate BIC weights in an analogous way to Akaike weights. The BIC weight of model $i$, denoted by $w_i$, is computed as

$$w_i = \frac{e^{-\Delta BIC_i/2}}{\sum_{i=1}^{Q} e^{-\Delta BIC_i/2}} \tag{7}$$

where $\Delta BIC_i$ is the difference between the BIC score of model $i$ and of the best model (i.e., the one with lowest BIC score), and $Q$ is the number of models.

## Relative deaths difference

To quantify the effect of vaccination, NPIs, or behavioural relaxation on deaths, we compute the relative deaths difference (RDD) as the relative difference between the total number of deaths as simulated by the original model and in a counterfactual scenario where vaccinations, NPIs, or behavioural relaxation are removed. The relative deaths difference is calculated as:

$$RDD = \frac{D_{counterfactual} - D_{original}}{D_{counterfactual}} \times 100\% \tag{8}$$

where $D_{original}$ and $D_{counterfactual}$ are the total number of deaths simulated in the original model and in the counterfactual scenario, respectively. The same approach, applied to infections, is used to compute the relative different of infections (RDI).

## Supporting information

**S1 Text. Supplementary analyses for the models and results.** In this supplementary file (PDF), we present additional analyses and results of our work.
(PDF)

## Acknowledgments

All authors thank the High Performance Computing facilities at Queen Mary University of London.

## Author contributions

**Conceptualization:** Yuhan Li, Nicolò Gozzi, Nicola Perra.

**Data curation:** Yuhan Li.

**Formal analysis:** Yuhan Li.

**Investigation:** Yuhan Li, Nicolò Gozzi, Nicola Perra.

**Methodology:** Yuhan Li, Nicolò Gozzi, Nicola Perra.

**Project administration:** Nicola Perra.

**Supervision:** Nicola Perra.

**Validation:** Nicola Perra.

**Visualization:** Yuhan Li.

**Writing – original draft:** Yuhan Li, Nicola Perra.

**Writing – review & editing:** Yuhan Li, Nicolò Gozzi, Nicola Perra.

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
