## [Decision Letter · Decision Letter 0]

26 Mar 2025

PCOMPBIOL-D-25-00068

Estimating behavioural relaxation induced by COVID-19 vaccines in the first months of their rollout

PLOS Computational Biology

Dear Dr. Perra,

Thank you for submitting your manuscript to PLOS Computational Biology. After careful consideration, we feel that it has merit but does not fully meet PLOS Computational Biology's publication criteria as it currently stands. Therefore, we invite you to submit a revised version of the manuscript that addresses the points raised during the review process.

Please submit your revised manuscript within 60 days May 26 2025 11:59PM. If you will need more time than this to complete your revisions, please reply to this message or contact the journal office at ploscompbiol@plos.org. Please include the following items when submitting your revised manuscript:

We look forward to receiving your revised manuscript.

Kind regards,

Brittany Rife Magalis, Ph.D

Academic Editor

PLOS Computational Biology

Roger Kouyos

Section Editor

PLOS Computational Biology

**Journal Requirements:**

3) Please amend your detailed Financial Disclosure statement. This is published with the article. It must therefore be completed in full sentences and contain the exact wording you wish to be published.

2) If any authors received a salary from any of your funders, please state which authors and which funders..

**Reviewers' comments:**

Reviewer's Responses to Questions

**Comments to the Authors:**

**Please note that one of the reviews is uploaded as an attachment.**

Reviewer #1: Yes, the review is uploaded as an attachment

Reviewer #2: I enjoyed reading this paper. Its research question has a high relevance to the field of public health - demosntration of impact of NPIs and vaccinations on reducing the burden of the pandemic and validation of assumptions about the effects of behaviors on epidemic dynamics. The tools used are relevant to the research question and were explained in sufficient detail. Separately, I would like to commend the lucidity of the narrative. I have only minor comments and recommend this work for publication.

1. One substantial but minor comment: in the methods, you mention that when NPIs are in place, while contacts outside of home are generally decreasing, you chose not to model the resulting increase in home contacts. I think it would be nice to include the discussion of how your conclusions about the effect of behaviour relaxation in the wake of vaccination rollout are affected by this assumption.

2. Further minor points, mostly cosmetic:

a. Frequently used 'firm minority'. I was not entirely sure about the meaning of this qualifier, propose to revise.

3. frequently 'however' is not placed correctly, disrupting the sentence flaw, making comprehension more difficult. For instance, the first paragraph of Introduction: 'Especially in the first months of their rollout, however, vaccination efforts 4 have encountered many challenges due to logistical issues, limited stockpiles and 5 healthcare capacity, vaccine nationalism, and spotty acceptance [7].' i propose 'However, especially in the first months of their rollout, vaccination efforts..' This comment is applicable to several instances in the text.

4. The same paragraph - the introduction opens with description of the roll our in the Global North - can you explain why? Was it the first location which started the vaccination rollout? And why is the worldwide rollout resctricted only to the Global North?

5. Same paragraph: I was not clear on what is 'Vaccine nationalism'. Does this refer to vaccine stockpiling?

6. Same paragraph, sentence 'The insufficient vaccination coverage in many areas proved inadequate

to prevent subsequent waves, thus leading to increased disease burden and to the

implementation of additional interventions to curb the spread of SARS-CoV-2.' Increased disease burden compared to what? Did you mean resurgence of new cases?

Overall, the first paragraph has narrative flows: we keep moving back and forth between the reasons why vaccination rollout was not smooth, and the consequences of it.

Finally, the last sentence of the paragraph, I would argue that the prime goal of vaccines was to reduce morbidity and mortality, with the hope that the same would follow for the transmission.

7. Occasional incorrect usage of 'how'. Illustration: the second paragraph of Introduction: 'The results from these efforts suggest how behavioural relaxation could reduce the positive gains brought about by vaccines thus leading to higher disease burden.' Here 'that' would be more accurate. I overall counted at least 3 similar instances.

8. Third paragraph of the introduction, lines 46-48. What do you mean by simpler baseline? A model without behavioral component? I would say so directly.

9. Line 53 'instead' is unnecessary. Frequently, 'instead' is not used correctly, hampering the flow of sentences.

10. Paragraph starting on line 58 almost belongs entirely in the discussion.

11. Line 72 - citation is needed.

12. Lines 132-133 Subsection Behavioural relaxation models, section Results . In this model, what drives individuals to become compliant or non-compliant? Is it other people holding these views? or is the term describing changes in compliant population given thus S^{c}' = \alpha S - gamma S_{c}?

13. Same paragraph, lines 141-142. In [24], only un-vaccinated people participated in compliance process. So in this sence, [24] is closer to models [1] and [2].

Results

1. Lines 169-179 - what is meant by 'less intense'? rate of growth, magnitude, cumulative number?

2. Lines 194-196 - Perhaps I am misunderstanding something, but would not the increase to levels beyond the pandemic levels result in value greater than 1 - but on the panel it is below 1?

3. lines 364-365. I am not clear, if you remove relaxation mechanisms, what stays to make the models different from baseline? especially in the case where the rate of transition to and from compliance does not depend on epidemiological and immunological state of the population.

Methods

1. line 459 'follow' should be 'follows'

2. line 462 'as' should be 'to be'

3. line 464 'after' should 'subsequently'

4. Figure 5, in flow diagram what does subscript 'k' in 'lambda' regers to? Coincidentally 'lambda' does not seem to be defined, and we do not know the relative difference in ability to transmit of vaccinated vs unvaccinated individuals.

5. Line 522: when you say 'allocating there 1%', do you mean they are moved fom existing infections or injected extra? what is % refers to then?

6. Tabke 4: overall vaccine efficacy for Delta variant is missing.

**Have the authors made all data and (if applicable) computational code underlying the findings in their manuscript fully available?**

Reviewer #1: Yes

Reviewer #2: Yes

PLOS authors have the option to publish the peer review history of their article (what does this mean?). If published, this will include your full peer review and any attached files.

Reviewer #1: No

Reviewer #2: No

**Figure resubmission:**
---

## [Decision Letter · Decision Letter 1]

24 Jun 2025

Dear Dr. Perra,

We are pleased to inform you that your manuscript 'Estimating behavioural relaxation induced by COVID-19 vaccines in the first months of their rollout' has been provisionally accepted for publication in PLOS Computational Biology.

Best regards,

Brittany Rife Magalis, Ph.D

Academic Editor

PLOS Computational Biology

Roger Kouyos

Section Editor

PLOS Computational Biology

Reviewer's Responses to Questions

**Comments to the Authors:**

Reviewer #1: I thank the authors for revising their manuscript and acknowledge the considerable effort made in re-running all the simulations with a second version of the model describing a more detailed natural history (SLPAIR), now presented as a sensitivity analysis. The expanded methods section and the addition of the table have substantially improved the clarity of the text and the modelling assumptions. I also appreciated the more thorough discussion of limitations and the additions to the supplementary material. For such reasons, I recommend this manuscript for publication.

Reviewer #2: All comments appear to be addressed satisfactory.

**Have the authors made all data and (if applicable) computational code underlying the findings in their manuscript fully available?**

Reviewer #1: None

Reviewer #2: Yes

PLOS authors have the option to publish the peer review history of their article (what does this mean?). If published, this will include your full peer review and any attached files.

Reviewer #1: No

Reviewer #2: No

---

## [Editor Report · Acceptance letter]

PCOMPBIOL-D-25-00068R1

Estimating behavioural relaxation induced by COVID-19 vaccines in the first months of their rollout

Dear Dr Perra,

I am pleased to inform you that your manuscript has been formally accepted for publication in PLOS Computational Biology. Your manuscript is now with our production department and you will be notified of the publication date in due course.

With kind regards,

Anita Estes
